



# Groundwater / meltwater interaction in proglacial aquifers

Brighid É Ó Dochartaigh[1], Alan M. MacDonald[1], Andrew R. Black[2], Jez Everest[1], Paul Wilson[3], W. George Darling[4], Lee Jones[5], Mike Raines[5]

1 British Geological Survey, Lyell Centre, Research Avenue South, Edinburgh EH14 4AP, United Kingdom
5 2 University of Dundee, School of Social Sciences, Dundee DD1 4HN, United Kingdom
3 Geological Survey of Northern Ireland, Dundonald House, Upper Newtownards Road, Belfast BT4 3SB, United Kingdom
British Geological Survey, Maclean Building, Wallingford OX10 8BB, United Kingdom
British Geological Survey, Environmental Science Centre, Keyworth NG12 5GG, United Kingdom

*Correspondence to*: Alan M. MacDonald (amm@bgs.ac.uk)

**Abstract.**

Groundwater plays a significant role in glacial hydrology and can buffer changes to the timing and magnitude of meltwater
flows. However, proglacial aquifer characteristics or groundwater dynamics in glacial catchments are rarely studied directly.
We provide direct evidence of proglacial groundwater storage, and quantify multi-year groundwater-meltwater dynamics,
through intensive and high resolution monitoring of the proglacial system of a rapidly retreating glacier, Virkisjökull, in SE
Iceland. Proglacial unconsolidated glaciofluvial sediments comprise a highly permeable aquifer in which groundwater flow in
the shallowest 20 – 40 m of the aquifer is equivalent to 4.5% (2.6-5.8%) of mean annual meltwater river flow, and 9.7% (5.8-
12.3%) of winter flow. Groundwater flow through the entire aquifer thickness represents 9.8% (3.6 – 21%) of annual meltwater
flow. Groundwater in the aquifer is actively recharged by local precipitation, both rainfall and snowmelt, and strongly
influenced by individual precipitation events. Significant glacial meltwater influence on groundwater within the aquifer occurs
in a 50–500 m river zone within which there are complex groundwater / meltwater exchanges. Stable isotopes, groundwater
dynamics and temperature data demonstrate active recharge from river losses, especially in the summer melt season, with more
than 25% of groundwater in this part of the aquifer sourced from meltwater. Such proglacial aquifers are common globally,
and future changes in glacier coverage and precipitation are likely to increase the significance of groundwater storage within
them. The scale of proglacial groundwater flow and storage has important implications for measuring meltwater flux, for
predicting future river flows, and for providing strategic water supplies in de-glaciating catchments.

## 1 Introduction

A major challenge in modern hydrology is predicting changes in freshwater flows and storage resulting from glacier retreat in
response to climate change (Jiménez Cisneros et al., 2014). Most glaciers worldwide have been in retreat since the mid-19th
century, with the loss of global glacier ice accelerating during the 21st century (Zemp et al., 2015). This change has the
potential to affect over one billion people who live in catchments where glacier melt contributes to river flow (Kundzewicz et



al., 2008). Glacial retreat is expected to increase meltwater river flows until the mid-late 21st century (Jiménez Cisneros et al., 2014; Lutz et al., 2014). Longer term, as glacier ice loss continues, meltwater flows will decrease (Jiménez Cisneros et al., 2014). This lessening of the role of glaciers in regulating flows will change the nature of glacier-fed rivers and the importance of other water sources in glacier catchments: rainfall, snowmelt and groundwater. Predicted impacts include changes to: the

frequency and magnitude of flooding (Jiménez Cisneros et al., 2014); hydroelectric power production (Laghari, 2013); drinking water and irrigation (Kundzewicz et al., 2008); ecosystem functioning of catchments (Brown et al., 2007); and groundwater recharge (Taylor et al., 2013).

The role of groundwater storage in the hydrology of deglaciating catchments is recognized, but to date there has been little direct hydrogeological investigation of groundwater-meltwater interactions (Levy et al., 2015, Vincent et al., 2018) with calls

for more (Heckmann et al. 2016, Vincent et al, 2018).  Indirect studies, inferred from river flow, indicate groundwater in Himalayan glacial catchments may be a significant source of delayed discharge to rivers (Andermann et al., 2012). Modelling of Himalayan catchments suggests that increased glacial melt this century will increase groundwater recharge from glacier runoff and the groundwater baseflow component in river flow (Immerzeel et al., 2013). Meltwater rivers in Alaska can potentially lose half their annual flow to groundwater (Liljedahl et al., 2017). Groundwater can comprise 15–75% of winter

river flows in glacial catchments in the European Alps, Peruvian Andes and Iceland (Bury et al., 2011; Hood et al., 2006; Malard et al., 1999; McKenzie et al., 2014; MacDonald et al., 2016). Direct experimental studies of groundwater in glacial environments are rare (Vincent et al. 2018): e.g. subglacial groundwater behaviour (Sigurðsson, 1990; Boulton et al., 2001; Boulton et al., 2007a, 2007b); groundwater flow in relict rock glaciers (Winkler et al., 2016); and the behaviour of shallow (<3 m) groundwater in glacial outwash plains in Iceland (Robinson et al., 2008; Robinson, et al., 2009a; Robinson, et al.,

2009b). The latter Icelandic studies demonstrated meltwater recharge to proglacial aquifers and linked retreating glaciers with declining groundwater levels (Levy et al., 2015).

In this study, we directly investigate the 3D aquifer properties of a proglacial floodplain (referred to here as *sandur*) of the Virkisjökull glacier in SE Iceland, to 15 m depth, using geophysics, drilling and hydraulic conductivity testing; and provide continuous time series data for groundwater, river stage/flow and precipitation over three years, with campaign sampling for

stable isotopes.  We explore the relationships between groundwater, meltwater flows and precipitation, revealing, seasonal and spatial hydrological patterns.

Iceland provides an ideal observatory for studying groundwater in deglaciating catchments. Ice melt from glaciers, which cover ~11% of Iceland, provides an estimated third of total river runoff (Björnson & Pálsson, 2008), but glacier retreat across Iceland (Sigurðsson et al, 2007) is forecast to produce significant changes in glacial catchment hydrology (Aðalgeirsdóttir et

al., 2011). The British Geological Survey, in collaboration with Veðurstofa Íslands (the Icelandic Meteorological Office), have studied the Virkisjökull catchment since 2009, monitoring rapid glacier retreat (Bradwell et al., 2013), retreat mechanisms (Phillips et al., 2013; Phillips et al., 2014), and researching glacial meltwater hydrology (MacDonald et al., 2016; Flett et al., 2017; Mackay et al., 2018).



## 2 Methodology

### 2.1 Study site

Virkisjökull is an outlet glacier of the Vatnajökull ice cap in SE Iceland (Figure 1), within the Virkisá river basin, which has a catchment area of ~32.5 km² to the confluence with the Svinafellsá river (MacDonald et al., 2016). Virkisjökull drains ice

steeply southwestwards from an elevation of >1800 m on the ice cap summit to <150 m at its terminus, with an average gradient of approximately 0.25. It has a high mass balance gradient, with net annual accumulation of more than 7 m w.e. a$^{-1}$ (metres of water equivalent per annum) at the ice cap summit (Guðmundsson, 2000) and net annual ice melt of more than 8 m w.e. a$^{-1}$ in the main ablation zone (Flett, 2016). The equilibrium line altitude on Virkisjökull is approximately 1150 m (MacDonald et al., 2016). The glacier has been retreating since 1990 (Hannesdóttir et al., 2015), with a marked acceleration in retreat rates since

2005 (Bradwell et al., 2013), during which time the glacier terminus has retreated by ~1 km and there has been extensive surface lowering.

The Virkisá river emerges from a small, shallow proglacial lake that has formed during the recent rapid deglaciation, and flows initially for 1 km over bedrock, flanked by moraines, and then for 4 km across the Virkisjökull sandur to the Svinafellsá river (Figure 1). The river drains glacial meltwater and virtually all precipitation falling on Virkisjökull glacier, adjacent hillslopes

and proglacial moraines. It occupies a single channel across the upper sandur, separating into a number of distinct channels across the lower sandur (Figure 1). The mean summer river flow over three years of continuous monitoring (2011–2014) ranged from 5.3–7.9 m³ s$^{-1}$; and significant river flow occurred in winter (mean 1.6–2.4 m³ s$^{-1}$). Isotopic studies (MacDonald et al., 2016), validated by numerical modelling (Mackay et al., 2018), demonstrate that summer river flows are governed by glacier ice melt, and that winter flows are a combination of meltwater, local precipitation and groundwater flow. The

Virkisjökull sandur falls from 100 to 50 m asl with a surface gradient of 0.017 (Figure 1). Over much of the sandur where river channels are actively migrating, there is little vegetation cover and no soil development. In more stable areas thin soils and more developed vegetation cover occur (Figure 1).

The proglacial area has a maritime climate with cool summers (mean summer air temperature 8–12 °C) and mild winters (1 °C). Air temperature in the Virkisá basin is controlled mainly by altitude, with an average annual lapse rate of -5 °C km$^{-1}$

(Flett, 2016; Mackay et al., 2018). Mean annual precipitation southwest of the Vatnajökull ice cap, including the Virkisjökull sandur, is ~1800 mm;, precipitation on the eastern side of the ice cap averages 3000 mm a$^{-1}$, and can exceed 7000 mm a$^{-1}$ on the ice cap summit (Guðmundsson, 2000). The proglacial area receives ~150 precipitation days per year, estimated from interpretation of three years of daily photographs (MacDonald et al., 2016), which also show that snow cover, even in winter, rarely lasts for more than a week before melting. Potential evapotranspiration over the sandur was estimated at ~450 mm a$^{-1}$

by Einarsson (1972) and actual evapotranspiration at 100–414 mm a$^{-1}$ by Jónsdóttir (2008).



## 2.1 Aquifer characterisation

Eight boreholes were drilled into the sandur to 9–15 m depth, in three transects approximately perpendicular to the river along a 3 km longitudinal reach in the upper, middle and lower study catchment (Figure 1a). Sediment samples collected during drilling were lithologically logged. The boreholes were installed as piezometers, with 88 mm diameter uPVC plain casing to

at least 5–12 m depth and a 3–6 m length of 0.5 mm slotted well screen below this (Table S1). A further two boreholes were drilled into volcanic bedrock, to 5.5 and 13.75 m depth, between the glacier terminus and the upper edge of the sandur (Figure 1a). Three methods were used to establish the physical aquifer properties of the sandur: (1) infiltration tests to 0.15 m depth at 20 locations, using a Guelph permeameter, and saturated hydraulic conductivity calculated by the Laplace method (Reynolds et al., 1983) (Table S2); (2) particle size analysis on 42 sandur sediment samples to 0.5 m depth, at 22 locations, and hydraulic

conductivity estimated using a modified Hazen formula suitable for heterogeneous glacial deposits (MacDonald et al., 2012; Williams et al. 2019) (Table S3); and (3) constant rate pumping tests of between 3.5 and 6 hours in each sandur piezometer, at rates of 0.5–1.8 l s$^{-1}$, and transmissivity estimated by the Jacob time-drawdown method (Table S1).

To measure aquifer thickness and depth to bedrock, two Tromino® passive seismic surveys were undertaken transversely across the Virkisjökull sandur, and a third longitudinally down the Svinafellsandur aquifer 4.5 km to the west, using a single

broad-band three-component seismometer with one vertical and two horizontal components. Measurements were recorded for 15 minutes at 50–100 m lateral intervals and data processed to derive depth to bedrock assuming typical shear wave velocities of 400–600 m s$^{-1}$ for Icelandic glacial sands and gravels (Bessason and Kaynia, 2002; Castellaro et al., 2005).  These data were interpreted with a previous seismic reflection survey in the area to infer sediment thickness and potential layering (Guðmundsson, 2002).


## 2.2 Groundwater monitoring and sampling

Monitoring of groundwater levels and temperature in sandur piezometers, at 15 minutes intervals, was undertaken from August 2012–May 2015 (34 months) using In-Situ Inc. Rugged Troll 100 non-vented pressure transducers at 7–8.4 m depth. Two In-Situ Rugged Barometer Trolls measured air pressure and temperature. River stage and discharge data are available for August

2012-May 2015 from an automatic stream gauge at Virkisá bridge (Figure 1) with two water-level sensors, checked using daily photographs and continuous flow measurements from a radar mounted beneath a bridge (MacDonald et al. 2016). From April 2013–March 2015 river stage and temperature were additionally monitored continuously every 15 minutes adjacent to piezometer U1 by an In-Situ Inc. Rugged Troll 100 pressure transducer (Figure 1). Rainfall data and temperature for the proglacial area were measured at the closest of the three Automatic Weather Stations installed by BGS in the catchment

(AWS1; 156 m asl). These weather stations were not equipped to measure snowfall, but daily photographs enabled periods of snowfall to be estimated.  Long term weather data from the Fagurhólsmýri weather station operated by the Icelandic





Meteorological Office (IMO) approximately 12 km south of the study site, and national scale gridded products (Nawri et al. 2017), were used to check the plausibility of weather data measured on site.

Hierarchical cluster analysis of groundwater level data was carried out on the entire dataset. Data were treated using the Standardized Groundwater level Index (Bloomfield and Marchant, 2013), which indicated the optimal number of clusters is

four. Groundwater flow was estimated assuming a mean aquifer width of 1 km, aquifer thickness at the river gauge from the passive seismic interpretation, average measured groundwater level gradient of 0.016 and hydraulic conductivity from median of all measured values (n = 64). Uncertainty was calculated from the interquartile range of measured K and uncertainty in aquifer thickness interpretation. The hydraulic conductivity of the deeper, unmeasured, sandur aquifer layer were estimated using the formula of MacDonald et al. (2012) taking into account a change in sediment state from very loose, to loose and firm

which is likely to over-estimate the reduction in pore space due to loading (Shmidt and McDonald, 1979). The total volume of groundwater stored in the aquifer was estimated using a conservative estimate of average aquifer porosity of 15% (Parrieux & Nicoud, 1990).

### 2.3 Groundwater isotopic sampling and analysis

Physico-chemical analysis and modelling were based on samples of groundwater from piezometers and springs collected

during three summer campaigns in September 2012, 2013 and 2014 and three winter (pre-melt) campaigns in January 2013, April 2013 and May 2014. Groundwater sampling was carried out after piezometers were purged by low-flow pumping until stable readings were obtained for field-measured parameters. Field measurements of specific electrical conductance (SEC), temperature and bicarbonate alkalinity by titration pH (Table S4), and of dissolved oxygen and redox potential (Eh), were made at the time of sampling. Samples for stable isotopes $\delta^{18}O$ and $\delta^{2}H$ were collected unfiltered in glass or Nalgene™

polyethelyne bottles and analyzed at BGS laboratories by isotope ratio measurement on a VG-Micromass Optima mass spectrometer. Data are quoted in permil (‰) with respect to Vienna Standard Mean Ocean Water (VSMOW) (IAEA/WMO, 2016); measurement precision was ±0.1‰ for $\delta^{18}O$ and ±1.0‰ for $\delta^{2}H$. Local precipitation stable isotope composition and a local meteoric line were estimated from International Atomic Energy Agency station data for Reykjavik (IAEA/WMO, 2016), supported by estimates for southeast Iceland (Arnason, 1977) and for south Iceland (Sveinbjörnsdóttir et al., 1995), as described

in MacDonald et al. (2016). The isotopic composition of meltwater in the Virkisá river was established by nine summer (melt) or winter (pre-melt) sampling campaigns from September 2011–December 2014 (MacDonald et. al., 2016).

The high topographic gradient of the catchment, with large climatic differences between the upland glacial accumulation area (>1800 m asl) and the lowland temperate proglacial area (<150 m asl), results in two easily distinguished isotopic compositions: for meltwater, and for precipitation across the proglacial area. A binary mixing model for $\delta^{2}H$ was applied to

investigate the relative contributions of local precipitation and of river water (which is dominated by glacial melt) to sandur groundwater, based on a two-component mixing equation. The end members applied for $\delta^{2}H$ composition were -76.1‰ for river water (Table S4) and -58.5‰ for average annual local precipitation (MacDonald et al., 2016). The fraction of local precipitation in sandur groundwater (FGW) was calculated using the formula FGW = $(\delta^{2}H_R - \delta^{2}H_P) / (\delta^{2}H_{GW} - \delta^{2}H_{HR})$, where





$\delta^2H_R$ is the composition of river water; $\delta^2H_P$ is the composition of local precipitation; and $\delta^2H_{GW}$ is the composition of sampled groundwater.

## 3 Results

### 3.1 Sandur structure and aquifer properties

The groundwater study catchment covers 6 km$^2$ and encompasses the sandur, adjacent hillslopes and moraines, and river outflow from the proglacial lake (Figure 1). Geophysical evidence from the passive seismic and previous seismic reflection survey indicates that (Figure 2, Figure S1) depth to bedrock increases from approximately 60–100 m in the upper sandur to 100–150 m in the lower sandur. The shallow aquifer material comprises loosely consolidated, moderately to poorly sorted, dominantly medium- to coarse-grained glaciofluvial sand, gravel and cobbles (Figure 2). All the sediment is of volcanic origin

and has been transported and deposited by the Virkisá river. The deeper deposits are not exposed, but nearby seismic interpretation confirms that the material is generally uniform to >50 m, reflecting the similar sediment derivation and deposition mechanisms (Guðmundsson, 2002). Although not directly observed in the seismic data there is a possibility that at greater depth (> 50 m) there exists more consolidated Pleistocene aged sediments which have been compaction by ice loading during earlier glaciations (Guðmundsson, 2002). Observations of bedrock from nearby exposures and two boreholes drilled

in bedrock reveal relatively massive and poorly fractured volcanic rock.

    The sandur aquifer is highly permeable to at least 15 m depth, with a median hydraulic conductivity of 35 m d$^{-1}$ (IQ range 25–40 m d$^{-1}$) (Figure 2a, Tables S2, S3). Transmissivity of the upper 15 m is 100–2500 m$^2$ d$^{-1}$ with median value of 600 m$^2$ d$^{-1}$ consistent with hydraulic conductivity measurements (Table S4). The permeability of the deeper sandur aquifer was not directly measured. However given the grain size distribution is the same as the shallow aquifer, and assuming the worst case

of compaction due to burial and ice loading (Shmidt and McDonald, 1979), median hydraulic conductivity may have reduced to 15 m d$^{-1}$ or at a worst case 6 m d$^{-1}$ (MacDonald et al., 2012). By contrast, the underlying bedrock has very low transmissivity below that which could be measured using a constant rate test (transmissivity < 0.25 m$^2$ d$^{-1}$). The sandur aquifer is unconfined. Depth to groundwater ranges from 0 to 4.4 m below ground level and maximum measured seasonal groundwater level fluctuations are 1.0–3.6 m. From 1 km down-sandur from its upper edge, there is extensive groundwater discharge at the

ground surface via perennial and ephemeral springs (Figure 2). A conservative estimate of the volume of groundwater stored in the full thickness of the aquifer is 51 ±15 million m$^3$, approximately 1 – 2 % of estimated ice volume in the glacier (Mackay et al. 2018).

### 3.2 Groundwater dynamics

    Groundwater level elevation falls from upper to lower sandur, with a gradient of 0.018 across the upper and 0.013 across the

lower sandur (Figure 2). In the upper sandur, closest to the glacier, groundwater levels adjacent to the meltwater channel are on average 1 m below river stage for most or all of the year (Figure 3a,b), leading to a strong piezometric gradient from river



to groundwater. Across the middle sandur, groundwater levels close to the river vary from 0.5 m below to 0.5 m above adjacent river stage, leading to complex meltwater/groundwater interactions. Here, piezometric gradients are generally from river to aquifer in the summer melt season, when river flows are highest; and from aquifer to river in winter, driven by high winter precipitation and associated recharge. From 2 km down-sandur, groundwater levels are above adjacent river stage for much of

the year, creating a piezometric gradient that drives visible groundwater discharge through seeps and springs to the river (Figure 2d) and ephemeral and perennial springs (Figure 2e).

Hierarchical cluster analysis of groundwater level data indicates two patterns of groundwater level fluctuation (Figure 3c): one driven primarily by local precipitation; and the second driven partly by precipitation but also strongly influenced by river stage, especially in summer (Figure 3d). Groundwater levels showing the first pattern (in piezometers U2, M1, M2, M3 and L3)

fluctuate dominantly in response to individual precipitation events and longer term precipitation patterns. The magnitude of groundwater level fluctuations typically increases with distance from the river. Rainfall is higher than its long term average throughout most of the winter and lower in summer, and this is generally reflected in the groundwater level fluctuations (Figure 3d). Groundwater levels showing the second pattern (in piezometers U1, L1 and L2) fluctuate in response to river stage as well as local precipitation, at seasonal (Figure 3d) and also at event timescales (Figure 3a). River stage is typically higher than

its long term average during peak summer melt, and groundwater levels in this group also remain close to or higher than their long term average throughout the summer (Figure 3d). The strongest response to river stage at a seasonal timescale is in piezometer L1, where groundwater levels during the 2013 summer melt season remained consistently higher than throughout the three winters from 2012–2014 (Figure 3d). The strongest response at an event timescale is in piezometer U1, where groundwater levels show consistent diurnal fluctuations during the summer melt season that coincide with diurnal melt-

controlled fluctuations in river stage (Figure 3a).

Piezometers U1 and U2 (20 m and 90 m from the river, respectively) illustrate the relative impacts of summer meltwater flows and large winter precipitation events on groundwater level–river stage gradients (Figure 3). In summer, low precipitation and large meltwater flows cause groundwater levels in U1 to rise above U2, creating a piezometric gradient away from the river (Figure 3a). During individual winter rain storms, groundwater levels in U2 rise higher than in U1, creating a piezometric

gradient towards the river (Figure 3b) and driving baseflow to the river further downstream in the middle sandur.

Mean estimated annual groundwater flow through the shallow part of the aquifer (20 – 40 m thick) is 0.19 m$^3$ s$^{-1}$ (0.093–0.30 m$^3$ s$^{-1}$), equivalent to 4.5% (2.7 – 5.8%) of mean annual river flow and 9.7% (5.8 – 12%) of mean winter river flow. The relatively small seasonal variation in groundwater levels means there is no significant seasonal variation in estimated groundwater flow across the aquifer. Overall groundwater flow through the total depth of the sandur aquifer is estimated as

0.42 m$^3$ s$^{-1}$ (0.12 – 1.1 m$^3$ s$^{-1}$) equivalent to 9.8% (3.6 – 22%) of mean annual river flow and 21% (7.7 – 46%) of mean winter river flow.



### 3.3 Stable isotopes and temperature

Stable isotope composition ($\delta^2H$ and $\delta^{18}O$) in groundwater from piezometers and springs was compared to that of river water and local rainfall (Figure 4, Table S4). Previous studies have demonstrated that meltwater and local rainfall on the proglacial area are easily distinguished using $\delta^2H$ and $\delta^{18}O$ due to the high elevation of the accumulation area (MacDonald et al. 2016).

Groundwater stable isotope compositions vary considerably across the sandur, spanning the range of compositions expected from meltwater and local precipitation (Figure 4). Piezometers (U1, L1, L2) identified from their hydrographs as most influenced by the river have isotopic compositions similar to river water, while piezometers whose hydrographs are influenced more by precipitation have a much wider range of isotopics composition, with U2 and M3 similar to local rainfall, and M2, M1 and L3 a mixture between local rainfall and river water. The springs showed a wide variety of compositions.

A binary mixing model developed for $\delta^2H$ indicates the relative proportion of precipitation and meltwater in groundwater (Figure 4b, 4c) and demonstrates a clear relationship with distance from the meltwater river. Within a zone extending up to 50 m from the river in the upper sandur, 130 m in the central sandur and 500 m in the lower sandur, groundwater in piezometers generally comprises more than 50% river water. Shallower groundwater from springs within this river zone is more influenced by local precipitation, but still comprises more than 25% meltwater. Beyond this zone, groundwater from both piezometers 15 and springs consistently comprises less than 25% river water (Figure 4c). Selected hydrochemical tracers and water temperature also help distinguish these two zones (Table S4). Specific electrical conductance (SEC) and bicarbonate ($HCO_3$) are significantly lower in those piezometers strongly influenced by the river than those where precipitation influence is dominant (Figure S1). River water temperature is relatively constant year-round at an average of 1.7˚C, and mean annual groundwater temperature is lowest in piezometers close to the river, and highest in those furthest from the river (Table S4).


## 4 Discussion

This study in Iceland shows that proglacial floodplains can form thick, highly permeable aquifers. By directly quantifying aquifer parameters and groundwater–meltwater interaction we have provided evidence of the significance of groundwater in proglacial hydrology. This has important implications for measuring meltwater flux, for predicting future river flows and ecological impacts, and for water supplies in de-glaciating catchments. Similar thick proglacial glaciofluvial aquifers with 25 high permeability and storage occur in other active glacial environments: e.g. elsewhere in Iceland (Robinson et al., 2008); the European Alps (Parrieux & Nicoud, 1990); and the Peruvian Andes (McKenzie et al., 2014), and with rapid deglaciation occurring globally proglacial aquifers are developing in many other locations increasing the importance of characterising groundwater (Vincent et al. 2018).




## 4.1 Groundwater flow

Our study shows that significant meltwater can flow through a glacierized catchment as groundwater, despite groundwater representing only a small proportion of the volume of water stored in glacial ice in the catchment. Reliable measurements of meltwater are important for calibrating cryospheric-hydrological models (Bliss et al., 2014; Lutz et al., 2014; Mackay et al.,
2018).  The estimated volume of groundwater flow through the shallowest $20 – 40$ m of the Virkisjökull proglacial aquifer is significant, $0.19$ m$^3$ s$^{-1}$, equivalent to approximately 4.5% of mean annual river flow or 9.7% of mean winter river flow, with estimates 9.8% and 21% respectively if flow through the full thickness of the aquifer is considered. Other studies in Iceland have proposed that a similarly large proportion of meltwater ($0.5 – 1$ m w.e. a$^{-1}$) can flow through the groundwater system, either from sub-glacial or proglacial recharge (Sigurdsson, 1990; Hemmings et al., 2016); meltwater river losses to
groundwater of up to 50% have also been reported (Liljedahl et al., 2017).  Measuring river flow in catchments with active glaciers is notoriously difficult, given the harsh conditions, the actively changing river beds, and the wide range in flows and sediment load. Therefore measurements are subject to high uncertainty. Here, we demonstrate that groundwater adds another source of uncertainty.  Measurements of river flows that rely solely on river stage in the proglacial area are likely to underestimate total annual meltwater flows, with much higher relative errors at low flows.

## 4.2 Meltwater / groundwater interaction

Groundwater-meltwater interactions are controlled by relative differences in water levels between the river and the proglacial aquifer, and vary both spatially, down the catchment, and seasonally. There is year-round active recharge of river water to the aquifer in the upper catchment, complex interaction in the middle of the sandur, and extensive groundwater baseflow to the
river and springs across the lower catchment. Distinct patterns of groundwater / meltwater dynamics are observed in groundwater level fluctuations and in groundwater stable isotopic composition, temperature and chemistry. In a zone extending up to 50–500 m from the meltwater river, the influence of the river on groundwater overshadows that of local precipitation. Here, recharge of meltwater to the aquifer from river losses has a significant impact on the physical, chemical and stable isotopic characteristics of groundwater in the proglacial aquifer.  The aquifer provides additional water storage, and
groundwater discharges back to the river further downstream through a large number of springs and seeps (Figures 1 and 4). This is consistent with other studies in glacierized catchments, which inferred groundwater baseflow to rivers of 15–75% (Bury et al., 2011; Hood et al., 2006; Malard et al., 1999; MacDonald et al., 2016; McKenzie et al., 2014).

However, away from the river the aquifer is recharged dominantly from local precipitation.  Active precipitation recharge to the aquifer is evident from groundwater stable isotopic composition and groundwater level response to precipitation, and
reflects high annual precipitation (rainfall and snow), high aquifer permeability, and low evapotranspiration linked to limited soil development and vegetation cover.  Recharge is likely to occur not only from direct precipitation on the sandur surface, but from ephemeral streams draining from hillslopes and groundwater seepage from surrounding moraines. Groundwater



discharge via springs and baseflow in the lower catchment supports surface water flows and local ecosystems and comprises groundwater derived mainly from local precipitation Figure 4).

Looking forward, as the glacier continues to melt, the proglacial aquifer will continue to have a buffering effect on river flow. High river flows will recharge the aquifer, whether caused by glacier icemelt, snowmelt or winter storms, as occurs in relic glacial outwash aquifers now in now-temperate areas (e.g. MacDonald et al., 2014), and will sustain springs, baseflow and surface ecosystems further down the catchment. Local precipitation falling on the aquifer is likely to continue to be a major source of aquifer recharge and contribution to river baseflow in addition to the groundwater discharging from other glacial deposits emerging within the landscape (MacDonald et al., 2016). In upland areas in northern Europe where glaciofluvial deposits from past glaciations are present, detailed studies have demonstrated that groundwater often comprises more than 50% of flow to river headwaters (Soulsby et al. 2005; Blumstock et al., 2015; Scheliga et al. 2017). Therefore, as glaciers continue to melt, groundwater baseflow is likely to become an increasingly important proportion of river flow in deglaciating catchments.

### 4.3 Proglacial aquifers as strategic water resources

This study has demonstrated that the Virkisjökull sandur is a highly productive aquifer with regular recharge. Similar thick proglacial glaciofluvial aquifers occur throughout the world, and are increasing in extent as glaciers recede, and are likely to also have the potential to sustain high quality reliable water supplies. In formerly glaciated areas, these aquifers are often targeted for public water supply (e.g. Ó Dochartaigh et al. 2015) because of their ability to sustain high yielding boreholes, their connectivity with rivers that provides additional recharge, and the generally high chemical quality of the groundwater compared to surface water. If projected glacier losses and increased precipitation in glacierized catchments are realized (Jiménez Cisneros et al., 2014), proglacial aquifers, recharged by local precipitation, represent a potentially significant store of high quality water in regions around the world that currently rely on glacier melt for water supply.

### Conclusions

Three years of investigations of groundwater and meltwater at Virkisjökull, SE Iceland, have enabled the aquifer parameters of the proglacial floodplain to be reliably characterised, and seasonal groundwater-meltwater dynamics to be quantified. The key findings from the research are:

1. Direct measurements of aquifer characteristics show consistently high permeability (35 m d$^{-1}$ n=64, IQR 25 – 40 m d$^{-1}$), and volume of groundwater storage (50 ±15 million m$^3$). The proglacial floodplain therefore forms a highly productive aquifer.

2. Significant meltwater flows through the shallowest 20 – 40 m of the proglacial floodplain as groundwater (0.19 m$^3$ s$^{-1}$. IQR 0.09-0.29 m$^3$ s$^{-1}$) equivalent to 4.5% of mean annual meltwater flow and 9.7% of mean winter flow. If the full




thickness of the aquifer is considered then flows of 0.42 m$^3$ s$^{-1}$ (0.12 – 1.1 m$^3$ s$^{-1}$) equivalent to 9.8% (3.6 – 22%) of mean annual river flow and 21% (7.7 – 46%) of mean winter river flow are possible.

3. Groundwater is recharged both from the meltwater river water and local precipitation falling on the aquifer, or draining from nearby hillslopes. Meltwater is particularly important in a zone from 50–500 m from the river where meltwater can form >50% of the recharge.

4. There are complex but consistent river/groundwater interactions: in the upper sandur, closest to the glacier, the river loses to groundwater much of the year; in the middle sandur the river loses to the groundwater in the summer melt and gains from groundwater in the winter low flows; in the lower sandur groundwater provides baseflow to the river through springs and baseflow seeps.

Proglacial aquifers are common worldwide and increasing in extent with deglaciation. These findings, therefore, have wider implications for measuring meltwater flux, for predicting future river flows, and for water supplies in de-glaciating catchments. Effectively understanding and characterizing groundwater flows and storage in catchments with glaciers, and incorporating this in hydrological models, will strengthen our ability to predict and manage the hydrological and environmental impacts of accelerating glacier retreat.

## Acknowledgments

This research was funded by the BGS-NERC Earth Hazards and Observatories Directorate and is published with the permission of the Executive Director of the British Geological Survey (NERC). We thank VatnajökulsÞjóðgarður for permission to install monitoring equipment; Vatnsborun ehf for borehole drilling and installations; Icelandair for assistance with equipment transport; and Veðurstofa Íslands and the people of Svinafell for research support. Tim Heaton at NERC Stable Isotope Facility undertook stable isotope analysis; James Sorensen carried out cluster analysis; and Craig Woodward at BGS assisted with diagrams (all BGS).

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







**Figure 1: Virkisjökull study catchment. (a) Study area on Virkisjökull sandur, SE Iceland, encompassing 6 km² groundwater catchment originating at proglacial lake outlet. Hillshade model generated from LiDAR DEM © Veðurstofa Íslands, 2010. (b) Piezometer M1 on upper sandur near catchment edge, showing established sandur vegetation and in middle distance, area of moraines. (c) Virkisá river on lower sandur in summer melt season showing braided channels and active, unvegetated sandur surface.**





**Figure 2: Geometry, geology and hydrogeology of the sandur aquifer. (a) Hydraulic conductivity and summer groundwater level contours. Other legend as Figure 1. Hillshade model generated from LiDAR DEM © Veðurstofa Íslands, 2010; (b) schematic cross section of hydrogeology, showing location of piezometer transects, spring discharge area and indicative groundwater flow lines; (c) geological section through river bank showing heterogeneous glaciofluvial deposition; (d) perennial groundwater-fed stream on lower sandur, associated with extensive growth of mosses and other aquatic vegetation; (e) groundwater discharge to otherwise inactive river channel on lower sandur, flowing to active channel in distance.**



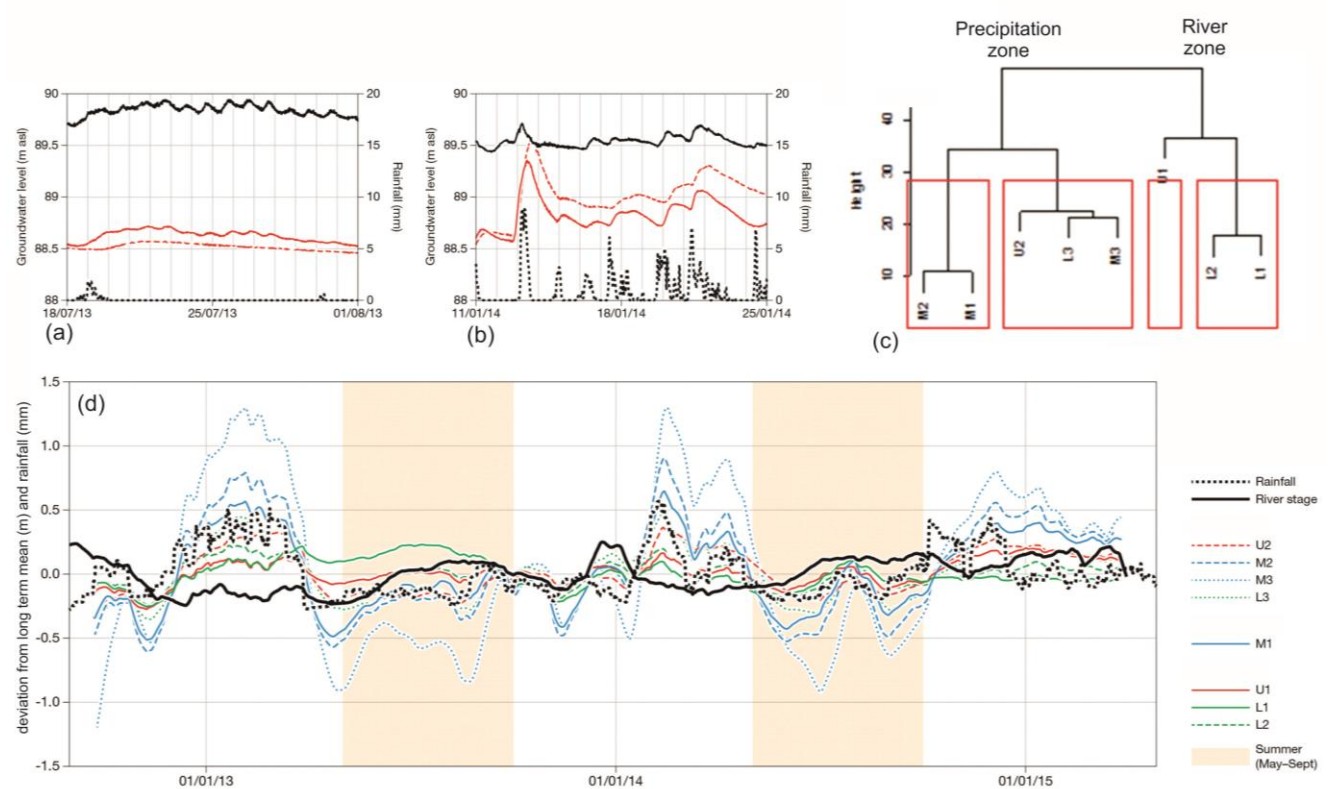

**Figure 3. Groundwater levels, river stage and precipitation. (a) Groundwater levels in upper sandur during a 14-day dry period in summer. (b) Groundwater levels in upper sandur during a 14 day rainy period in winter. Piezometer U1 (solid) is 20 m from the river; piezometer U2 (dashed) is 90 m from the river. (c) Dendogram obtained by hierarchical cluster analysis of groundwater level data from sandur piezometers (piezometer locations in Figure 1). The highest level break shows two clusters representing piezometers where groundwater is influenced dominantly by local precipitation (U2, M1, M2, M3, L3) and piezometers where groundwater is influenced by the meltwater river (U1, L1, L2). Red boxes show the four optimal sub-clusters indicated by data standardization. (d) Monthly running mean of river stage, hourly precipitation and groundwater level, as variation from long term average (LTA = 0).**





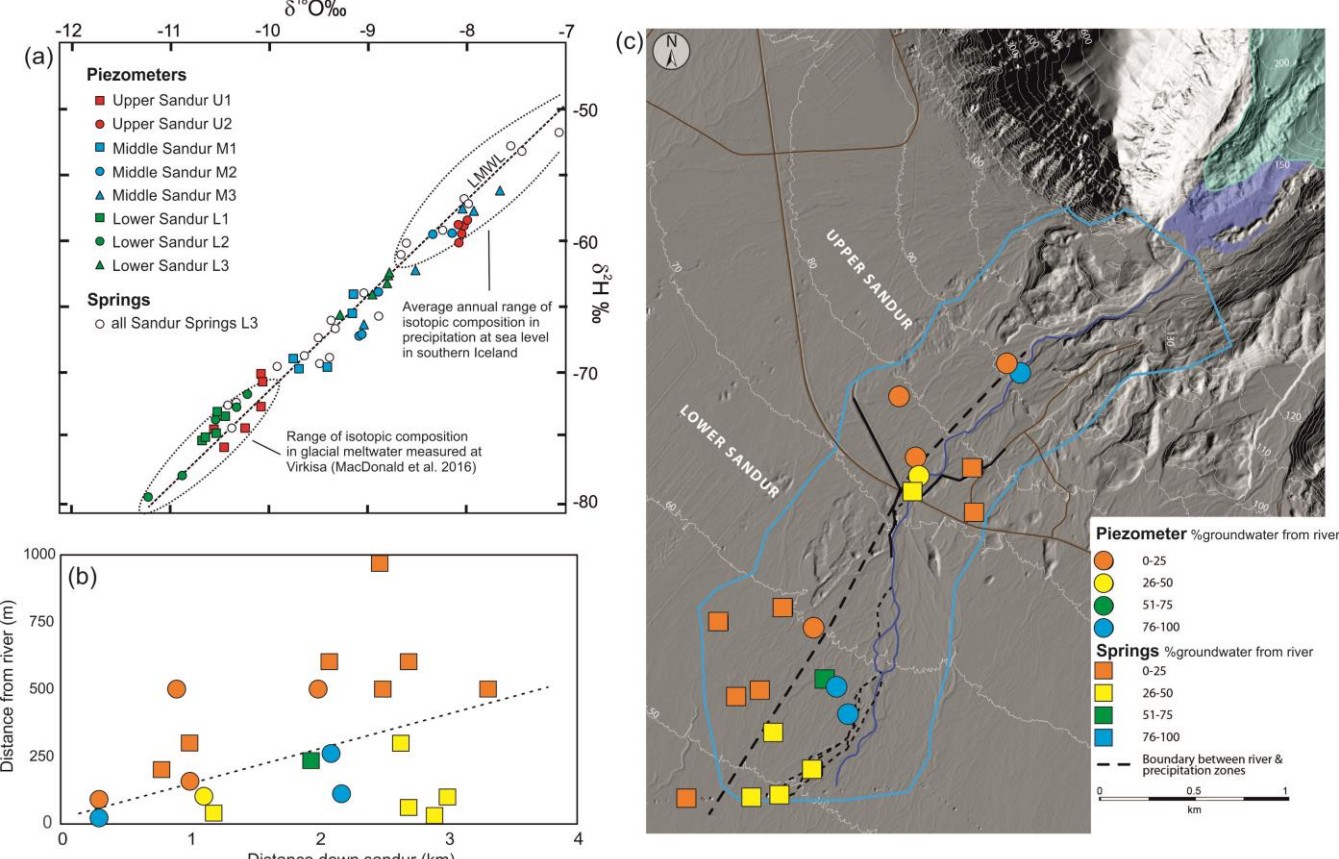

**Figure 4. Stable isotope composition of waters and results of binary mixing model of $\delta^2H$ in groundwater. (a) Stable isotope composition of sandur groundwater in piezometers and springs. Individual piezometers labelled; piezometer locations in Figure 1. Also shown are ranges in stable isotope composition of precipitation and river water (MacDonald et al. 2016). Plotted on Local Meteoric Water Line (LMWL) calculated for Reykjavik. (b) Plot of mean proportion of groundwater recharged from the river using binary mixing of model of $\delta^2H$ by perpendicular distance from the river and down-sandur. (c) Map of mean proportion of groundwater estimated to be recharged from river using binary mixing model for $\delta^2H$. Hillshade model generated from LiDAR DEM © Veðurstofa Íslands, 2010.**