# Peer review of "Groundwater - glacier meltwater interaction in proglacial aquifers"

_Hydrology and Earth System Sciences, 2019_

## Referee Comment (RC1) · Aude Vincent (Referee) · 22 May 2019

General comments

This paper addresses groundwater in a glaciarised catchment, an object still rarely considered, and even less through a multi-years and multi-technics study. And, as well argued in the paper, it is already, and will be even more in the coming years, of great interest for many areas, regarding several issues including water supply. Significant conclusions are reached, based on strong new data.

The scientific significance of the paper is thus excellent, and definitely within the scope of HESS.

It's scientific quality is high too, I have only few comments to reach full understanding

and potential reproduction of the results (see specific comments section). The method section is very clearly detailed.

The presentation is of excellent quality as well, minors corrections and suggestions are listed in the technical corrections section. Figures and references are exhaustive. Title and abstract are adequates.

Specific comments

- Shouldn't subsection 2-1 "Study site" be included in the introduction section rather than the Methodology section? And 2 subsections are labelled "2-1": the "Study site" one and the "Aquifer characterisation" one.

- section 3-1: line 5: Could you explicit what define the groundwater catchment boundaries? Except for one indication on Figure S1 there is no discussion of this.

- section 3-1: line 21: How do you know that the underlying bedrock has very low transmissivity? Thanks to tests in the 2 dedicated piezometers?

- section 3-2: lines 26-31: Could you explicit how you calculate the mean estimated annual groundwater flow through the shallow part of the aquifer and the total depth of the aquifer?

- Just to be sure I got this right: if there are tills under Virkisjökull glacier, they are not in continuity with the sandur downstream?

Technical corrections

- some of the 's-1' are not formatted properly (section 2-1 Study site mostly).

- section 2-1 Aquifer characterisation: "Within a zone extending up to 50 m...": use a non-breaking space to avoid line break between 50 and m

- section 2-1 Aquifer characterisation: lines 2 and 5: add the date the sandur and bedrock boreholes were drilled

- section 2-1 Aquifer characterisation: line 13: add the date the seismic survey were performed

- section 2-2: suggestion for the title: "Groundwater, Surface hydrology and Precipitation monitoring"

- section 2-2: line 22: . . . in the 8 sandur piezometers;

- section 2-2: line 24 line break after "temperature."

- section 2-2: line 28 line break after "(Figure 1)."

- section 2-3: suggestion for the title: "Groundwater sampling and analysis"

- section 3-1 line 18: is the table referred to really Table S4?

- section 3.3: "M1, M2 and L3" instead of "M2, M1 and L3"

- section 3.3 "Within a zone extending up to 50Âǎm...": use a non-breaking space to avoid line break between 50 and m

- maybe simplify the figures numberingÂǎ? To have some figures labelled ÂńÂǎ1, 2...ÂǎÂż and one ÂńÂǎS1ÂǎÂż is sometimes a bit confusing.

- Maybe add the piezometers drilling dates in the text and in table S1.

Why are the 3 piezometers in the bedrock not included in table S1 or in a specific table?

- Are the dates of the infiltration tests indicated in the text? It could also be indicated in table S2.

- same thing for the particle size analysis and table S3.

- add in table S4 the years of the sampling campaigns

―――――――――――――

---

## Referee Comment (RC2) · Anonymous Referee #2 · 25 May 2019

Review of Dochartaigh et al., "Groundwater / meltwater interaction in proglacial aquifers"

Although there is growing recognition of the importance of groundwater in glacierized watersheds, there have been relatively few studies that directly characterize groundwater in such systems. This study serves to help fill that gap by using groundwater wells and isotope data to quantify groundwater storage, groundwater discharge, and the contribution of glacial meltwater to groundwater. While on their own, these methods are relatively straightforward, applying them in glacierized, mountainous settings can be challenging, and thus their findings about meltwater-groundwater interactions is a valuable contribution to our understanding of glacierized watersheds. This manuscript is overall well-written.

There are some aspects of the presentation that need clarification, however.

1. Clarify "meltwater". Ultimately, I believe the authors use the term "meltwater" to refer to glacier melt (not snowmelt), and they assume the river water consists of glacier melt. This was confusing, however. First of all, there are some references to "snowmelt", so I was unsure at times whether "meltwater" should also include "snowmelt". Also, the authors at times discuss groundwater/meltwater interactions after presenting results about river water-groundwater interactions, and it was not obvious that the reader is supposed to assume the river water and meltwater are treated as being the same (I pointed out specific lines below). I suggest the following. Be explicit about glacier meltwater (which could include snowmelt on the ice?) vs. local snowmelt. Also, be explicit about the assumption that the river water is glacier melt. However, I would caution against treating river water and meltwater as interchangeable, because the authors point out that the river water can consist of groundwater (during the wet season in middle elevations and all year in the lower elevations).

2. Clarify the isotope mixing model implementation. The methods section describes taking winter and summer water samples for isotope analysis, but no seasons are identified in the results. Isotope values can be very seasonally dependent – was this taken into account for the mixing model implementation? Also, what isotope value was used for the precip end-member? Was it the range of values indicated on Fig. 4 for precip at sea level? How well does isotopic value for precip at sea level apply to local precip on the mountain slope? Finally, and most importantly: why is the mixing model applied to estimate river contributions to groundwater in the middle and lower elevations (this is what Fig. 4c appears to show)? This contradicts elsewhere in the manuscript that describes flow to occur from groundwater to the river during the wet season in mid-elevations, and at all times in lower elevations.

3. Clarify the interpretation of comparing groundwater discharge to stream discharge. Your wording seems to imply that the groundwater discharge is all from glacier meltwater (even though it also includes recharge from local precip), and that stream discharge is all glacier meltwater (even though lower sections include groundwater). Perhaps this is not what is intended, but, for example, point 2 in the Conclusions makes it sound like the 0.19 m3/s groundwater discharge is meltwater. And the abstract mentions "meltwater river flow", implying that the river only consists of (glacier?) meltwater. I suggest rewording.

4. Clarify aquifer width. Explain the assumption of 1 km width – this is a strong assumption that controls your ultimate groundwater discharge estimate. Can you explain it – is it b/c it is the approximate width of the watershed, and you assume the groundwater-shed is similar? When you report your groundwater discharge result, you should be careful to note the uncertainty due to assuming this width.

Other minor comments:

- p. 1, Line 21-23: These two sentences are confusing.  I think the first sentence sets up the reader to expect that groundwater is mainly fed by local precip.  The second line could be edited to better emphasize that glacial meltwater is even more important than precip inputs at certain places.  Part of the confusion for me in the second line is that it was not evident that the river water is all meltwater, and so I did not realize that "groundwater / meltwater exchange" is actually groundwater /river water exchange, where river water is meltwater.

- would "groundwater-meltwater" be better than "groundwater/meltwater"?

- p.1, Line 25: be explicit that "meltwater" here is "glacier meltwater"

- p. 2, Lines 8-20.  I have a few other suggestions for your lit review.  Also examining a direct link between meltwater and groundwater, Saberi et al. 2019 used a watershed model to show that groundwater discharge increases by 20% with meltwater contributions in a glacierized watershed in Ecuador.  Harrington et al. 2018 found that 100% of winter streamflow originates from gw (rock glacier spring discharge) in the Canadian Rockies.  Baraer et al. 2015 is a nice summary paper about groundwater contributions to discharge in multiple glacierized watersheds in Peru.  Also, you cite Hood et al. 2006, but you did not mention catchments in the Canadian Rockies.

References:

Baraer, M., Mckenzie, J., Mark, B. G., Gordon, R., Bury, J., Condom, T., … Fortner, S. K. (2015). Contribution of groundwater to the outflow from ungauged glacierized catchments: A multi-site study in the tropical Cordillera Blanca, Peru. Hydrological Processes, 29(11), 2561–2581. https://doi.org/10.1002/hyp.10386

Harrington, J. S., Mozil, A., Hayashi, M., & Bentley, L. R. (2018). Groundwater flow and storage processes in an inactive rock glacier. Hydrological Processes, 32(20), 3070–3088. https://doi.org/10.1002/hyp.13248

Saberi, L., McLaughlin, R. T., Ng, G.-H. C., La Frenierre, J., Wickert, A. D., Baraer, M., … Mark, B. G. (2019). Multi-scale temporal variability in meltwater contributions in a tropical glacierized watershed. Hydrology and Earth System Sciences, 23(1), 405–425. https://doi.org/10.5194/hess-23-405-2019

- p. 2, Line 25: delete comma after "revealing"

- p. 3, Line 5: specify "m.a.s.l."  You specify this elsewhere, so be consistent.

- Fig. 1: helpful to be explicit in caption that your abbrevations are for Upper (U1-U2), Middle (M1-M3), and Lower (L1-L3).

- p. 3, Line 19: would be good to clarify that "meltwater" is glacier meltwater0

- p. 3, Line 26: typo ";,"

- p. 4, Line 12: comment on use of Jacob time-drawdown method for unconfined aquifer?  (If not in main text, then in supplementary info?)

- p. 4, Line 29: write out "BGS" in the first occurrence

- p. 5, Line 5: change "Groundwater flow" to "groundwater discharge"

-p. 5, Line 11: why was porosity not measured directly?

-p. 6, Line 13: "compacted" instead of "compaction"

-p. 7, Line 2: confusing to see "meltwater/groundwater interactions" here. You need to explicitly explain that you assume the river water to be meltwater.

-p. 7, Line 2: confusing to see "piezometric gradients are from river to aquifer". Flow is along negative gradient, so this phrase technically means that flow is from aquifer to river (I think you mean the opposite).

- Figure 3:
        - improve resolution
        - (a) and (b) needs legend
        - (d) y-axis label is confusing, why "and rainfall"? I believe all lines are "deviation from long term mean"
        - (d) color-code by precip vs. river impact groups? Would be easier to see what is described in text

-p. 7, Line 13: M1 is also very close to river. Any idea why it did not show up in 2nd pattern?

- Figure 4: Similar to comment for Fig. 3: I suggest color-coding to emphasize correspondence with main sources. Also, why does legend say "L3" after "All Springs"? Other figures show springs scattered, not just by L3.

- p. 8, Line 10: be explicit here that "meltwater" is assumed to be same as river water.

---

## Author Response (AR1)

Below is our responses to the reviewers in blue and then our changes to the document in red. Please see the marked up version of the document to see where the changes have been made. We thank the reviewers for such useful comments and think we have now made a much clearer manuscript

**Reviewer 1**

Thanks for the comments the time taken to go through the article:

**General Comments**

Thanks for the assessment of the high quality of the science, significance and presentation.

**Specific comments**

*- Shouldn't subsection 2-1 "Study site" be included in the introduction section rather than the Methodology section? And 2 subsections are labelled "2-1": the "Study site"one and the "Aquifer characterisation" one.*

Thanks for the comment. I find its generally 50/50 whether to put the explanation of the study site in the introduction or methods. Happy to put it into the Introduction if that makes it clearer

We have moved to the introduction and numbered it 1.1

*- section 3-1: line 5: Could you explicit what define the groundwater catchment boundaries? Except for one indication on Figure S1 there is no discussion of this.*

We have used the surface water catchment boundary as it extends from the defined glacial valley onto the sandur as an approximate indication of the groundwater catchment boundary. It is marked on Figure 1 – but I note that we have not put it in the legend – which we'll correct. Later in the paper – the SIs are much more powerful at defining the zone of interaction between the river and the groundwater – so the groundwater catchment is only used to help give an approximation of groundwater flow within the Sandur

Included a section in 1.1 to show how we estimated the groundwater catchment

*- section 3-1: line 21: How do you know that the underlying bedrock has very low transmissivity? Thanks to tests in the 2 dedicated piezometers?*

Yes – two piezometers drilled into the lavas – and the experience of local community in trying to get water supplies form the lava.

We have modified the sentence to make this clearer

*- section 3-2: lines 26-31: Could you explicit how you calculate the mean estimated annual groundwater flow through the shallow part of the aquifer and the total depth of the aquifer?*

The groundwater flow methodology is explained in the methods section (p 5 lines 5 – 10).  We use Darcy's equation and parameterise with the head – which we have measured, the permeability which is measured in the top 15 m of the aquifer, the width of the aquifer from the approximate groundwater catchment taken from the surface water catchment using dGPS.  The measurement of the total depth of the aquifer is discussed in page 5  (lines 13 – 19).  There is some limited evidence that the

5    aquifer may become more consolidated at depth – so we quote a flow through the shallow depths (<40 m) as well as through the full depth. We can modify the methods section to explicitly mention the darcy equation

We have repeated in the results section that we have used Darcy's equation.

*- Just to be sure I got this right: if there are tills under Virkisjökull glacier, they are not in continuity with the sandur downstream?*

10   That right – there is negligible direct contact between the glacier and the sandur – primarily because of a bedrock high between the glacier and the sandur

No changes made

**Technical sections**

15   Thanks – we will modify as suggested

See track changes version

**Reviewer 2**

20   Review of Dochartaigh et al., "Groundwater / meltwater interaction in proglacial aquifers"

*Although there is growing recognition of the importance of groundwater in glacierized watersheds, there have been relatively few studies that directly characterize groundwater in such systems. This study serves to help fill that gap by using groundwater wells and isotope data to quantify groundwater storage, groundwater discharge, and the contribution of glacial meltwater to groundwater.*

25   *While on their own, these methods are relatively straightforward, applying them in glacierized, mountainous settings can be challenging, and thus their findings about meltwater-groundwater interactions is a valuable contribution to our understanding of glacierized watersheds. This manuscript is overall well-written.*

We thank the reviewers for their comments and appreciate the time to carefully examine the document

*There are some aspects of the presentation that need clarification, however.*

30       1.   *Clarify "meltwater". Ultimately, I believe the authors use the term "meltwater" to refer to glacier melt (not snowmelt), and they assume the river water consists of glacier melt. This was confusing, however. First of all, there are some references to "snowmelt", so I was unsure at times whether "meltwater" should also include "snowmelt". Also, the authors at times discuss groundwater/meltwater interactions after presenting results about river water-groundwater interactions, and it was not obvious that the reader is supposed to assume the river water*

35       *and meltwater are treated as being the same (I  pointed out specific lines below). I suggest the following. Be explicit*

*about glacier meltwater (which could include snowmelt on the ice?) vs. local snowmelt. Also, be explicit about the assumption that the river water is glacier melt. However, I would caution against treating river water and meltwater as interchangeable, because the authors point out that the river water can consist of groundwater (during the wet season in middle elevations and all year in the lower elevations).*

Thanks for this observation of ambiguity in the language. We mean glacier meltwater (ice + snow on glacier) when we discuss meltwater, rather than low level snow melt on the sandur aquifer which melts quickly (within a few days or weeks) of winter precipitation events. We take your suggestion of referring to glacier meltwater throughout and defining this as ice + snowmelt.

We have referred to glacial meltwater generally – unless referring to the river directly

> 2. *Clarify the isotope mixing model implementation. The methods section describes taking winter and summer water samples for isotope analysis, but no seasons are identified in the results. Isotope values can be very seasonally dependent – was this taken into account for the mixing model implementation? Also, what isotope value was used for the precip end-member? Was it the range of values indicated on Fig. 4 for precip at sea level? How well does isotopic value for precip at sea level apply to local precip on the mountain slope? Finally, and most importantly: why is the mixing model applied to estimate river contributions to groundwater in the middle and lower elevations (this is what Fig. 4c appears to show)? This contradicts elsewhere in the manuscript that describes flow to occur from groundwater to the river during the wet season in mid-elevations, and at all times in lower elevations.*

**Seasonality**

Although seasonal samples were taken for groundwater there was no significant seasonal variation  See the means and standard deviations in Table S4.  This is because of the long residence times of in the aquifer >> 1year which integrates the seasonal cycle.

Two sentences have been added to the results section to explain that the seasonal variation is low – and point to the table in the supplementary material which demonstrates this

**End members**

The end points of glacial melt and weighted annual mean of local rainfall were used in the analysis, and the data and explanation behind this discussed in an earlier paper (MacDonald et al. 2016: 10.1017/aog.2016.22).  Here is a summary of this discussion. The glacier is an excellent location to carry out these studies as there is such a marked contrast between the stable isotope composition of the two end members -76.1 ± 2.6 ‰ $\delta 2H$, for glacier meltwater and -58.5 ± 6‰ $\delta 2H$, for rainfall.  The composition for rainfall was calculated from the weighted annual mean from the nearest IAEA station and compared to a two other published results from the snouts of glaciers at Öræfajökull– which all suggest an annual weighted mean for rainfall of approx. -58‰ Árnason (1977), Sveinbjörnsdóttir and others (1995). MacDonald et al. 2016: 10.1017/aog.2016.22 also collected samples from local shallow springs unaffected by the river and got similar results of 58.5  ± 6‰ for shallow groundwater- which integrates the annual rainfall signal.

The glacier meltwater endpoint is determine from the weighted mean of river water stable isotopes as the river leaves the small proglacier area before reaching the Sandur.  There is a small variability in the signature measured from both summer melt and winter melt.  -76.1 ± 2.6‰ $\delta 2H$.  Summer melts can be slightly more depleted (-77 – 78‰ $\delta 2H$) reflecting a higher component

of ice melt – For example a large survey of ice stable isotopes gave a mean of $-77.3 \pm 3.7$‰ $\delta 2H$ (MacDonald 2016). However this variability is negligible when comparing to the signature of local precipitation of -58.5 ± 6‰.

No further explanations are given in the text.  The reference to the earlier work of MacDonald 2016 should be sufficient for readers who are interested in more of the detail

5    *why is the mixing model applied to estimate river contributions to groundwater in the middle and lower elevations?*

The stable isotope signature of glacier meltwater contributions is determined from samples taken from the river as it leaves the glacier proglacial area – before entering the sandur with the complex surface water groundwater interactions.  Because of the permeability of the aquifer, much of the flow in the aquifer is sub parallel to the river and the main contribution of glacier meltwater to the  aquifer is likely to be just as the river enters the sandur.  The stable isotope composition of the river is not

10    monitored downstream for this particular study.  Therefore the assumptions of using the glacier meltwater stable isotope signature and local precipitation as endmembers for groundwater in the Sandur – however downstream – still holds true.

We have ensured that we refer to glacier meltwater as the end member here which should help avoid confusion of the potential evolution in river water SI composition as it travels down the Sandur.  We have also added a sentence to the methods section to clarify that glacier meltwater is used and that evolution of river for isotopes downs stream in summer is negligible and

15    insignificant when compared to the difference between river and local precipitation isotopic composition .

3.    *Clarify the interpretation of comparing groundwater discharge to stream discharge. Your wording seems to imply that the groundwater discharge is all from glacier meltwater (even though it also includes recharge from local precip), and that stream discharge is all glacier meltwater (even though lower sections include groundwater).*

20    *Perhaps this is not what is intended, but, for example, point 2 in the Conclusions makes it sound like the 0.19 m3/s groundwater discharge is meltwater. And the abstract mentions "meltwater river flow", implying that the river only consists of (glacier?) meltwater. I suggest rewording.*

Thanks for this.  It is certainly not our intention to suggest that groundwater is discharge is from meltwater only – quite the opposite.  We demonstrate the local precipitation is very important for groundwater recharge.

25    L21-22 in the abstract.  Groundwater in the aquifer is actively recharged by local precipitation, both rainfall and snowmelt, and strongly influenced by individual precipitation events

I assume its line 18 - 20 in the abstract that causes confusion?  Here we compare the groundwater flow to the river flow.

E.g. Line 20 Groundwater flow through the entire aquifer thickness represents 9.8% (3.6 – 21%) of annual meltwater.

We    suggest    we    alter    this    to    just    "river    flow"    and    delete    "represents".        So

Groundwater flow through the entire aquifer thickness, sourced both from local precipitation and glacier meltwater, is approximately 9.8% (3.6 – 21%) the magnitude of annual flow in the river

And then add in a sentence indicating that local precipitation remains the largest source of recharge to the aquifer, before the discussion of the extent of river water / groundwater interaction

These have been implemented in full with glacial meltwater being referred to and clarification throughout that that the groundwater flow is not all glacier meltwater.

4. *Clarify aquifer width. Explain the assumption of 1 km width – this is a strong assumption that controls your ultimate groundwater discharge estimate. Can you explain it – is it b/c it is the approximate width of the watershed, and you assume the groundwater-shed is similar? When you report your groundwater discharge result, you should be careful to note the uncertainty due to assuming this width.*

Yes the width is based on the hydrological boundary which was mapped on the ground with dGPS. The large uncertainties attributed to the flow at depth 9.8% (3.6 – 21%) reflects this uncertainty, although we believe to have reduced some of the uncertainty in the shallower groundwater system4.5% (2.6-5.8%)

Refer to Reviewer 1 – we have clarified the groundwater catchment. Uncertainties in groundwater flow are calculated from uncertainty in aquifer thickness and permeability which are likely to be much greater than potential uncertainties in aquifer width. Therefore we have left the indicative width as 1 km and included uncertainties in depth and permeability

*Other minor comments:*

*- p. 1, Line 21-23: These two sentences are confusing. I think the first sentence sets up the reader toexpect that groundwater is mainly fed by local precip. The second line could be edited to betteremphasize that glacial meltwater is even more important than precip inputs at certain places. Part of the confusion for me in the second line is that it was not evident that the river water is all meltwater, and so I did not realize that "groundwater / meltwater exchange" is actually groundwater /river water exchange, where river water is meltwater. - would "groundwater-meltwater" be better than "groundwater/meltwater"?*

As discussion above – yes we will clarify this and refer to glacier meltwater rather than meltwater. We will also change to groundwater / river exchanges here and throughout where we are discussing direct exchanges between the meltwater river and the groundwater

Thanks or pointing this out. We have reworded and made it much clearer that precipitation is more important for the overall aquifer.

*- p.1, Line 25: be explicit that "meltwater" here is "glacier meltwater"*

Thanks – will do

Done

*- p. 2, Lines 8-20. I have a few other suggestions for your lit review. Also examining a direct link between meltwater and groundwater, Saberi et al. 2019 used a watershed model to show that groundwater discharge increases by 20% with meltwater contributions in a glacierized watershed in Ecuador. Harrington et al. 2018 found that 100% of winter streamflow originates from gw (rock glacier spring discharge) in the Canadian Rockies. Baraer et al. 2015 is a nice summary paper about groundwater contributions to discharge in multiple glacierized watersheds in Peru. Also, you cite Hood et al. 2006, but you did not mention catchments in the Canadian Rockies.*

Thanks for the references – we will read and consider them

Included Saberi at al 2019 in our discussion which adds weight to our findings that groundwater flow can lead to an under estimation of glacier melt.

5  We've added Harrington to our examples of rock glacier groundwater flow – adding a Canadian example to that o the Alpine one given

Barear et al has been added as a good example of where groundwater contributes much of winter discharge

Thanks also for the detailed comments below which will help considerably tightening the manuscript

10  Answers to two more significant ones

*- p. 4, Line 12: comment on use of Jacob time-drawdown method for unconfined aquifer? (If not in main text, then in supplementary info?)*

Yes will put in the supplementary material.  And generally applies well if the drawdown is low compared to the saturated thickness of the aquifer

15  In the end we have clarified in the main text that we have modified for unconfined conditions and also used Theis recovery and added added a reference demonstrating that these methods are appropriate for unconfined conditions

*-p. 7, Line 13: M1 is also very close to river. Any idea why it did not show up in 2nd pattern?*

Yes – this puzzled us too.  We believe that the reason is probably because there is a small locally sourced channel close to the
20  piezometer – giving the local precipitation a stronger control on groundwater levels

We have also implemented in full the other editorial suggestions – see marked up document

[revised manuscript text omitted]

---

## Author Response (AR2)

**Response to editor**

As requested we have include the DOIs for the datasets used in the manuscript as we have just had them accepted by the

5  NGDC.

We have included them both in the data sources section at the end of the manuscript and also within the methods section where

we have referred to the datasources and put them in the reference list, as advised by the NGDC.

We have also picked up minor errors associated with punctuation and references.

10  I trust that all is in order.  Please get back in touch if there are any other queries.

Best wishes

Alan

[revised manuscript text omitted]